# Rapid Reversal of Carbapenemase-Producing *Pseudomonas aeruginosa* Epidemiology from *bla*_VIM_- to *bla*_NDM_-harbouring Isolates in a Greek Tertiary Care Hospital

**DOI:** 10.3390/antibiotics13080762

**Published:** 2024-08-12

**Authors:** Efthymia Protonotariou, Georgios Meletis, Nikoletta Vlachodimou, Andigoni Malousi, Areti Tychala, Charikleia Katsanou, Aikaterini Daviti, Paraskevi Mantzana, Lemonia Skoura

**Affiliations:** 1Department of Microbiology, AHEPA University Hospital, School of Medicine, Aristotle University of Thessaloniki, S. Kiriakidi Str. 1, 54636 Thessaloniki, Greece; meletisg@hotmail.com (G.M.); nikoletta.vlachodimou@gmail.com (N.V.); aretich@gmail.com (A.T.); katsanouh@gmail.com (C.K.); davitikaterina@hotmail.com (A.D.); vimantzana@gmail.com (P.M.); lemskour@auth.gr (L.S.); 2Laboratory of Biological Chemistry, School of Medicine, Aristotle University of Thessaloniki, 54124 Thessaloniki, Greece; andigoni@auth.gr

**Keywords:** *Pseudomonas aeruginosa*, carbapenemases, NDM, VIM, carbapenem resistance, hospital infections

## Abstract

Carbapenemase-producing *Pseudomonas aeruginosa* strains present a specific geographical distribution regarding the type of carbapenemase-encoding genes that they harbor. For more than twenty years, VIM-type enzymes were the only major carbapenemases that were detected among *P. aeruginosa* isolates in Greece until the emergence of NDM-1-encoding *P. aeruginosa* in early 2023. In the present study, we present the rapid reversal of the carbapenemase-producing *P. aeruginosa* epidemiology from *bla*_VIM_- to *bla*_NDM_-harbouring isolates that occurred in our hospital since then. Between January 2023 and February 2024, 139 isolates tested positive for carbapenemase production with the NG-Test CARBA 5 immunochromatographic assay. Eight isolates were processed with the Hybrispot antimicrobial resistance direct flow chip molecular assay, and the first NDM-producing isolate was further analyzed through whole genome sequencing and bioinformatics analysis. Multiple resistance genes were detected by molecular techniques in accordance with the extensively drug-resistant phenotype. The isolate that was subjected to whole-genome sequencing belonged to the *P. aeruginosa* high-risk clone ST308, and the *bla*_NDM_ was located in the chromosome in accordance with previously reported data. During the study period, NDM-producing isolates were increasingly detected, and only five months after their emergence, they overcame VIM producers. Our results indicate the potential of this new clone to spread rapidly and predominate within healthcare institutions, further restricting the already limited treatment options.

## 1. Introduction

Carbapenem-resistant *Pseudomonas aeruginosa* (CR-PA), a major pathogen worldwide, is listed as a high-priority pathogen by the World Health Organization (WHO) Bacterial Priority Pathogens List due to its high level of antibiotic resistance and the severe infections it causes, particularly in healthcare settings [1]. In Europe, large difference are observed in the percentages of CR-PA among countries, ranging from 5% to over 50. Regarding Greece, *P. aeruginosa*’s resistance is quite high for carbapenems, at 48.9% compared to the EU population-weighted mean of 16.5% [2].

Carbapenem resistance in *P. aeruginosa* is multi-factorial [3]. Different mechanisms may be involved and may even coexist in non-susceptible isolates, like the loss of the OprD porin, the presence of inducible AmpC type β-lactamases, the overexpression of efflux pumps, and the production of carbapenemases. Among them, carbapenemases are by far the most effective because they confer high-level resistance to all or almost all β-lactams, including ceftazidime, cefepime, piperacillin–tazobactam, and carbapenems. Moreover, since carbapenem-resistance genes are often located in mobile genetic elements, they can be easily disseminated between bacterial cells and species [4].

*P. aeruginosa* presents a remarkable ability to acquire and host carbapenemase-encoding genes and, interestingly, most of the clinically important carbapenemases that can be found in carbapenem-resistant Gram negatives were reported for the first time in *P. aeruginosa* isolates worldwide. Moreover, *P. aeruginosa* strains that encode specific carbapenemase genes, like IMP, KPC, VIM, and NDM, present a characteristic geographical distribution and are commonly related with the so-called high-risk clones [5].

NDM is closely related to Enterobacterales and Acinetobacter *spp*. and has been much less frequently detected in *P. aeruginosa* [6]. For many years, carbapenemase-producing *P. aeruginosa* in Greece were almost exclusively harboring genes encoding for VIM-type carbapenemases [7,8,9,10]. Recently, however, there has been a rapid reversal to NDM-encoding *P. aeruginosa* in our hospital soon after the first isolation in 2023, and such isolates are quickly overcoming the VIM producers.

The present study was designed to monitor the carbapenem-resistant *P. aeruginosa* epidemiology from February 2023 to February 2024 within our hospital. Additionally, whole genome sequence analysis was conducted on the first NDM-encoding isolate (91845) that was recovered in May 2023 from a blood culture.

## 2. Results

### 2.1. Carbapenem Resistant P. aeruginosa Isolates

During the study period, 211 carbapenem-resistant *P. aeruginosa* isolates were recovered from various hospital wards. Among them, 139 isolates originating from 73 male and 66 female patients were found to be positive for carbapenemase production.

### 2.2. Detection of Resistance Determinants by the NG-Test CARBA 5 and the Hybrispot Antimicrobial Resistance Direct Flow Chip (AMR)

Overall, the NG-Test CARBA 5 detected 72 VIM producers and 66 NDM producers. One isolate (214929) was found to be positive for both carbapenemases (Table 1).

There was a 100% concordance of the NG-Test CARBA 5 results with those reported by the Hybrispot. Moreover, Hybrispot revealed the presence of additional resistance determinants to other antibiotic classes in the eight isolates tested using this method (Table 2).

Interestingly, after their emergence in *P. aeruginosa* in our hospital, NDM-encoding genes have been increasingly detected. Moreover, NDM-producing isolates rapidly overcame the VIM producers within this species, as shown in Figure 1.

### 2.3. Genomic Characterization of NDM-1-Producing Isolate 91845

The NDM-1-producing isolate was genomically characterized for the presence of antimicrobial resistance genes, the presence of virulence genes, and MLST. In addition, the isolate was comparatively analyzed against the most closely related isolates, and the genomic origin of the *bla*_NDM-1_ gene, either chromosomal or plasmid, was investigated.

High-depth sequencing of isolate 91845 resulted in 155.95 M reads that were filtered to meet high-quality base calling criteria (Q30 for 96.62% of 150 bp reads). The assembled genome contains 92 contigs with an N50 value of 246.540 bp and a total genome size of 6,919,402 bp. The most closely related isolate, based on the genome-wide nucleotide identity, is a recently isolated clinical strain in Greece (GCA_030504675_1, 99.948% identity), followed by the reference *bla*_NDM-1_ human isolate from Singapore (CP020704, 99.93% identity). Two clinical isolates from the neighboring countries Italy (RefSeq assembly: GCF_013276295_1) and Serbia (RefSeq assembly: GCF_022559565.11) were identified as being closely related, with 98.99% and 98.93% identity similarity values, respectively.

### 2.4. MLST, Antimicrobial Resistance and Virulence Genes

Isolate 91845 belongs to the *P. aeruginosa* high-risk clone ST308 that has been previously reported to carry carbapenemase genes. The antimicrobial resistance analyses resulted in the following AMR genes: *aac(3)-Id, aac(6′)-Ib-cr, aac(6′)-Ib3, aac(6′)-Il, aadA11, aph(3″)-Ib, aph(3′)-IIb, aph(6)-Id*, *bla*_NDM-1_, *bla*_OXA-10_, *bla*_OXA-488_, *bla*_PAO_, *catB7, crpP, dfrB5, floR*, *fosA, msr(E), qacE, qnrVC1, rmtF, sul1,* and *sul2*. Ten virulence genes were identified: *xcpP, exoU, flgC, pchB, mbtH-like, lasI, pilG, pscS,* and *pscF, hcp1* (Table 3).

### 2.5. Genomic Origin of the bla_ndm_ Gene

*bla*_NDM-1_ was detected in a 2323 bp contig which also 0included an IS91 family transposase (Figure 2). No plasmid replicons have been detected in the assembled genome by PlasmidFinder, while PlasmidSPAdes did not detect *bla*_NDM-1_ in the 27 variable-length plasmid constructs. These results are in line with previous evidence that ST308 *P. aeruginosa* strains including *bla*_NDM-1_ are located in chromosomal regions. However, PLASme and MOB-suite classified the 2323 bp contig as a plasmid sequence originating from Enterobacterales. The prediction is considered as being high confidence, except for a 10 bp ambiguous regions identified by PLASme. In support of this finding, MOB-suite detected an AD312 plasmid (CP034849) originated from *Escherichia coli* that includes the *bla*_NDM-1_-containing contig as well as five other contigs of the same type (contigs 18, 39, 50, 58, and 105).

To further assess whether these AD312-type plasmid contigs resemble known plasmids, blastn searches against complete plasmids were applied, yet with no strong evidence of similar plasmids. To further investigate the potential chromosomal origin of the *bla*_NDM-1_ gene, genome-wide pairwise alignment analysis against the reference CP020704 (Singapore) genome was performed. The genomic region neighboring *bla*_NDM-1_ in CP020704 has strong homology with the disjoint contigs 50, 79, 80, and 105 of our isolate. Figure 2, showing the 100% blast-based similarity with chromosomal regions, further supports the chromosomal origin of the *bla*_NDM-1_ gene. Interestingly, the highly homologous contigs 50 and 105 have been also identified as AD312 plasmids by MOB-suite.

## 3. Discussion

The New Delhi metallo-β-lactamase was named after its first report back in 2008 from Sweden regarding a patient previously hospitalized in New Delhi, India [11]. Following this initial detection, it soon became obvious that NDM-1 was already widely disseminated in the Indian subcontinent as well as in other countries, although to a lesser extent [12,13,14]. NDMs are categorized as Ambler group B carbapenemases together with other clinically important metallo-enzymes, such as VIM and IMP. Metallo-β-lactamases bear zinc in their active center and commonly hydrolyze all β-lactams except aztreonam and cefiderocol, whereas they are not inhibited by the β-lactamase inhibitors or by boronic acid. NDMs, however, present some additional unique characteristics. They hydrolyze aztreonam and they commonly present negative modified Hodge test results [15]. Resistance to cefiderocol in *P. aeruginosa* is still rare, but for the time being it is related only to some NDM-1-encoding lineages [16]. In Greece, NDM-1-encoding genes were introduced in late 2011 [17] and, since then, they have been well-established together with KPC, VIM, and OXA-48 carbapenemases among Enterobacterales and especially *Klebsiella pneumoniae*, as shown also by recent data from our hospital [18,19].

*P. aeruginosa* is a species that is commonly related to carbapenemases including KPC [20] and most of the clinically important metallo-β-lactamases [21,22,23,24]. Interestingly, there is a clear geographical distribution of isolates harboring different enzymes around the world, with KPCs being mainly located in Latin America, VIM and IMP presenting a global distribution, and NDMs being mostly located in India and Australia [25]. This specific distribution is associated with the spread of specific high-risk clones [5]. In Europe, the first report of NDM-encoding *P. aeruginosa* was reported from Serbia [26]. In 2010, at the Military Medical Academy, seven carbapenem-resistant *P. aeruginosa* isolates were recovered. Molecular investigation proved that two of them were the first NDM-producing *P. aeruginosa* worldwide. The first NDM-positive isolate was recovered from a urine sample of a 61-year-old Serbian woman with an intra-abdominal abscess, while the second one was detected in a wound of a 63-year-old woman who underwent laparotomy for intestinal carcinoma. Both patients had no previous travel history and died shortly after their admission.

The first NDM-1 in *Pseudomonas aeruginosa* clinical isolates from India were reported between 2011–July 2012. A total of 4 out of 200 *P. aeruginosa* clinical isolates investigated were NDM producers. The first was isolated from a central venous catheter culture of a 66-year-old man with a necrotizing soft tissue infection of the left lower limb. The second was also isolated from a venous catheter culture and was detected in a male patient with scarring after gallbladder removal surgery. Colistin was used successfully for both patients. The third isolate was found in the urine sample of a 56-year-old man with pyelonephritis. In this case, the treatment included colistin and amikacin. The fourth isolate was detected in a pus sample from a male patient with a compound comminuted fracture of the tibia that developed a surgical site infection [27].

In mid-2012, a case of a patient from Slovakia with NDM-1 *P. aeruginosa* was reported [28]. The strain was isolated from a 44-year-old patient who was hospitalized in the intensive care unit of the University Hospital in Bratislava, due to acute respiratory failure. The patient’s condition worsened and respiratory support was necessary, as he developed pneumonia. The causative agent of ventilator-associated pneumonia was *Acinetobacter baumannii*, according to the result of a bronchoalveolar lavage culture. During his hospitalization in the ICU, NDM-producing *P. aeruginosa* and *Enterococcus faecalis* were isolated from his blood culture. Ιn 2012, NDM-producing *P. aeruginosa* was also found in France. This strain was detected in a urine culture of a 63-year-old woman who was admitted to the military hospital in Bégin because of complicated pyelonephritis. It is noteworthy that the patient was hospitalized in Serbia three months before her admission [29]. A few months later, in late May 2013, the first NDM-producing *P. aeruginosa* was reported from Italy [30]. A 40-year-old man diagnosed with acute lymphoblastic leukemia in first remission was admitted to a hematology unit in Rome in order to undergo stem cell transplantation. Interestingly, the patient had been hospitalized in December 2012 in Belgrad, Serbia. Fifteen days after transplantation, the patient was febrile and neutropenic, and the blood cultures taken on that day were positive for *P. aeruginosa*. Two days later, the patient rapidly deteriorated and was admitted to the intensive care unit, where he died a few hours later due to septic shock. The isolate was afterwards found to be an NDM producer. The first autochthonous Italian NDM-encoding *P. aeruginosa* was isolated in August 2019. At that time, a 77-year-old female patient, who lived in a rehabilitation center, was admitted to Bari’s hospital because she suffered from chronic respiratory failure. During her hospitalization, she developed a urinary tract infection, and *P. aeruginosa* was isolated in the urine culture. The patient did not receive any antibiotics, and three days later she became febrile. At that point, *P. aeruginosa* was also isolated in her blood cultures. The patient’s condition deteriorated rapidly, and she finally died of sepsis in November 2019 [31].

After their emergence in Greece [32] and for more than 20 years, carbapenemase-producing *P. aeruginosa* harbored exclusively VIM-type enzymes, most often those of the VIM-2 family [33]. In 2023, NDM-producing *P. aeruginosa* were reported for the first time from Larisa [34]. The first isolate was recovered on 16 May 2023 from the bronchial secretions of a female patient with previous hospitalizations in various Greek ICUs before her admission at the University Hospital of Thessaly in April 2023. Eight more cases with *P. aeruginosa* producing NDM carbapenemases were detected in that Hospital during the same time period. There are many important similarities between our characterized isolate 91845 and those reported from Larisa. First, all characterized isolates in Greece belong to the epidemic high-risk international clone ST308. Second, they all present a high identity percentage with the reference *bla*_NDM-1_ human isolate from Singapore and, third, the NDM-encoding gene seems to be located in the chromosome.

In the past, another case of rapid reversal in carbapenemase epidemiology for yet another species, namely *K. pneumoniae*, has also occurred in Greece. More specific, carbapenem-resistant *Klebisella pneumoniae* KPC changed mostly to the VIM type rapidly after the introduction of ceftazidime/avibactam into clinical practice as a novel agent that effectively inhibited the action of KPC enzymes [35]. In *P. aeruginosa*, the reason is clearly not the introduction of a new antimicrobial and may rather be attributed to the ST308 potential for rapid spread.

The predominance of NDM-producing versus VIM-producing *P. aeruginosa* is worsening the already difficult situation regarding the treatment of the respective infections. VIM-type enzymes hydrolyze almost all β-lactams except aztreonam and cefiderocol, whereas NDM-type enzymes hydrolyze β-lactams including aztreonam. Even though cefiderocol is not yet introduced in Greece, the spread of NDM-producing *P. aeruginosa* could influence its success rates a priori. Indeed, resistance to cefiderocol in *P. aeruginosa* has been observed up to now only among some NDM-producing isolates. Ceftolozane/tazobactam, ceftazidime/avibactam, imipenem/relebactam, and meropenem/vaborbactam are not active against both VIM and NDM carbapenemases because the metallo-β-lactamases are not inactivated by the novel β-lactamase inhibitors. When active, amikacin and fosfomycin could be used to treat effectively urinary tract infections [36,37]. In our hospital epidemiology, however, resistance to these drugs among carbapenemase-producing *P. aeruginosa* is common. Thus, not surprisingly, genes encoding for aminoglycoside-modifying enzymes and fosfomycin resistance have been detected by the Hybrispot and NGS methods in our study. Polymyxin resistance is rare for carbapenemase-producing *P. aeruginosa*, and this applies in our case as well. Our isolates are susceptible to colistin by both Vitek2 and the broth microdilution method. Colistin, however, is a formerly abandoned antibiotic that presents nephrotoxicity issues, and its actual clinical usefulness has been widely questioned [38].

Our data suggest that the departments that were first affected by NDM-producing *P. aeruginosa* were, in chronological order, the internal medicine department A, ophthalmology, neurosurgery, and surgery department B. These departments are located far from one another inside the hospital; nevertheless, connections between them (by moving personnel for, e.g., physiotherapy) can neither be excluded nor verified retrospectively. Additionally, our institution is a tertiary care hospital admitting patients from numerous acute- and long-term care facilities, and as our colleagues from Larisa, Greece, suggested, NDM-producing *P. aeruginosa* was already circulating in Northern Greece [34]. That being said, we expect to have had multiple index patients; therefore, it would not seem safe to draw conclusions upon intrahospital dissemination.

The present study has some limitations that should be noted. Due to economic restrictions, only one of the study’s isolates was sequenced. The molecular characterization of all isolates would have shed more light on the dynamics of the *bla*_NDM_ spread among *P. aeruginosa* in our hospital. More molecular data analysis could evidence a probable clonal expansion and maybe provide more information about the interactions and interplay between VIM-producing and NDM-producing strains, since the first NDM + VIM producer has already been recovered in our hospital (isolate 214929). Depending on how the present situation may develop, this could be an interesting issue for future research.

## 4. Materials and Methods

### 4.1. Hospital Setting and Patient Data

The study was conducted in AHEPA University Hospital, which is located in Northern Greece and has a 700-bed capacity. All isolates included in this study were collected as part of the standard of care protocol. Data of the patients were retrieved from the hospital’s electronic database.

### 4.2. Study Sample

All carbapenem-resistant (resistant to both imipenem and meropenem) *P. aeruginosa* isolates recovered from all hospital wards between January 2023 and February 2024 were tested for carbapenemase production with the NG-Test CARBA 5 (NG-Biotech Laboratoires, Guipry-Messac, France) immunochromatography assay. In cases of multiple isolations per patient, only one isolate was included in the study. Moreover, the results of the first NDM-1 producer (91845), the first NDM and VIM producer (214929), and of six more isolates presenting weak lines in immunochromatography were confirmed by the HybriSpot antimicrobial resistance direct flow chip (AMR) (Máster Diagnóstica, Granada, Spain) molecular assay. Additionally, the first NDM-1-producing *P. aeruginosa* (91845) isolated from a blood culture was subjected to whole-genome sequencing (WGS) and bioinformatic analysis.

### 4.3. Bacterial Identification and Susceptibility Testing

Bacterial identification was carried out by matrix-assisted laser desorption/ionization time-of-flight mass spectrometry technology (Bruker Daltonics, Bremen, Germany), and antimicrobial susceptibility testing was performed using the Vitek2 system (bioMérieux, Marcy l’Étoile, France). Additionally, confirmatory susceptibility testing was accomplished, where applicable, by using the MICRONAUT-S MDR MRGN-screening system (Bruker Daltonics GmbH & Co. KG, Bremen, Germany). For antimicrobial susceptibility results interpretation purposes, the European Committee on Antimicrobial Susceptibility Testing (EUCAST) 2023 clinical breakpoints (v 13.0) were applied.

### 4.4. NG-Test CARBA 5

The NG-Test CARBA 5 (NG-Biotech Laboratoires, Guipry-Messac, France) is a rapid visual multiplex immunochromatographic assay for the qualitative detection and differentiation of five major carbapenemases from carbapenem-resistant bacterial colonies. The assay consists of one cassette that includes specific areas for the detection of KPC, OXA-48-like, VIM, IMP, and NDM enzymes together with a specific control (C) area. The assay was performed according to the manufacturer’s instructions, and results were interpreted visually at 15 min after incubation in room temperature.

### 4.5. Hybrispot Antimicrobial Resistance Direct Flow Chip (AMR)

The Hybrispot antimicrobial resistance direct flow chip (AMR) (Máster Diagnóstica, Granada, Spain) is a microarray-based assay for the in vitro detection of multiple antibiotic resistance genes. More specifically, the assay allows the simultaneous detection of 54 antibiotic resistance genetic markers associated with multi-drug-resistant organisms, such as carbapenem-resistant Gram-negative bacteria, extended-spectrum beta-lactamase (ESBL) producers, vancomycin-resistant Enterococci (VRE), and methicillin-resistant *Staphylococcus aureus* (MRSA). The assay’s markers represent the major gene families including SHV, CTX-M, GES, SME, KPC, NMC/IMI, SIM, GIM, SPM, NDM, VIM, IMP, OXA23-like, OXA24-like, OXA48-like, OXA51-like, OXA58-like, MecA, VanA, and VanB. Additionally, AMR is able to identify *P. aeruginosa, Acinetobacter baumannii, E. coli, Klebsiella pneumoniae*, and *S. aureus*.

### 4.6. Whole-Genome Sequencing and Bioinformatics Analysis

Whole-genome sequencing was performed with theDNBSEQ-G99 high-throughput Sequencing Set based on MGI’s core DNBSEQ^TM^ sequencing technology (MGI Tech, Shenzhen, China). Raw sequencing reads were quality-checked and trimmed using fastp [39]. The draft genome was assembled using skesa 2.5.1 [40].

The antimicrobial resistant determinants, the sequence type, and plasmid replicons were detected by Staramr version 0.9.1 [41], which integrates diverse molecular profiling tools including CGE’s multilocus sequence typing [42], Resfinder [43], and PlasmidFinder [44]. To further determine whether a contig originates from a chromosome or a plasmid source of the *bla*_NDM_-containing contig, plasmid prediction and construction tools, such as PlasmidSPAdes [45], PLASme [46], and MOB-suite [47], were applied. LASTZ [48] identified highly homologous sequences against reference bla-NDM-1 assembly from the reference Singapore isolate and constructed the pairwise alignment mappings between the two isolates using Proksee [49]. Plasmid sequences were further annotated based on the presence of AMR genes by CARD Resistance Gene Identifier v.5.2.1 [50] and protein families that are linked to the integration/excision, replication/recombination/repair, transfer, and stability/transfer/defense processes of the plasmids bymobileOG-db [51]. FastANI [52] was used to calculate the genome-wide average nucleotide identity against closely related *bla*_NDM_-containing *P. aeruginosa* strains.

## 5. Conclusions

An ongoing spread of NDM-producing *P. aeruginosa* is taking place in Greece, where VIM-encoding *P. aeruginosa* strains are already endemic. The intensification of the existing infection control policies seems to be the only realistic measure to tackle the widespread dissemination of yet another potential threat to public health.

## Figures and Tables

**Figure 1 antibiotics-13-00762-f001:**
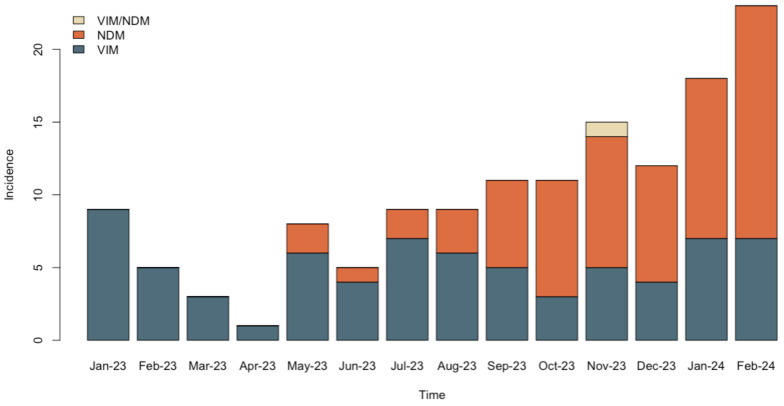
Reversal of the carbapenemase-producing *P. aeruginosa* epidemiology from VIM to NDM in the hospital setting.

**Figure 2 antibiotics-13-00762-f002:**
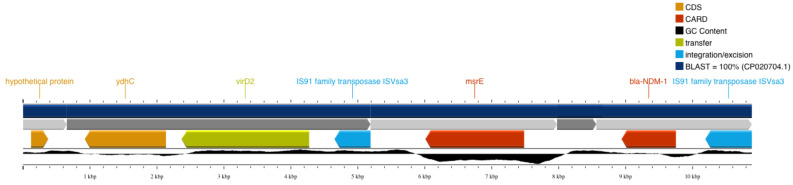
The 2323 bp contig including *bla*NDM-1 and the neighboring IS91 family transposase.

**Table 1 antibiotics-13-00762-t001:** Carbapenemase-producing *P. aeruginosa* isolates included in the study. VIM: Verona integron-encoded metallo-β-lactamase; NDM: New Delhi metallo-β-lactamase; BAL: bronchoalveolar lavage; CVC: central venous catheter ICU A: intensive care unit A; INT B: internal medicine department B; NB: neurology department B; INT A: internal medicine department A; BH: hematology department; CARD: cardiology department; SURGC: surgery department C; SURGB: surgery department B; NS: neurosurgery; EMERG: emergency department; OPHT: ophthalmology department; NEPHR: nephrology department; CARDSURG: cardiosurgery.

Month/Year	Isolate ID	Date of Isolation	Carbapenemase	Specimen Type	Ward	Age (Years)	Patient Gender
January 2023	401	2 January 2023	VIM	Rectal surveillance swab	ICU A	99	M
	371	2 January 2023	VIM	Urine	NS	68	M
	496	2 January 2023	VIM	Urine	SURGB	79	M
	2342	4 January 2023	VIM	Urine	NB	61	F
	11102	17 January 2023	VIM	Blood	ICU A	68	F
	12955	20 January 2023	VIM	Blood	INT A	83	M
	14753	23 January 2023	VIM	Rectal surveillance swab	ICU A	77	F
	15011	23 January 2023	VIM	Sputum	INT B	82	M
	20359	30 January 2023	VIM	Blood	SURGB	58	M
February 2023	34945	20 February 2023	VIM	Urine	SURGC	45	F
	34955	20 February 2023	VIM	BAL	ICU A	72	F
	38980	24 February 2023	VIM	Blood	NS	81	M
	41488	28 February 2023	VIM	Wound	EMERG	70	M
	41273	28 February 2023	VIM	Blood	EMERG	87	F
March 2023	47005	8 March 2023	VIM	BAL	SURGB	41	M
	47930	8 March 2023	VIM	Urine	EMERG	57	M
	52508	14 March 2023	VIM	Urine	NB	74	F
April 2023	84353	27 April 2023	VIM	Blood	INT A	87	M
May 2023	86305	1 May 2023	VIM	Blood	INT A	65	M
	91845	8 May 2023	NDM	Blood	INT A	85	M
	96242	15 May 2023	VIM	BAL	ICU A	64	F
	99556	19 May 2023	VIM	Blood	CARD	76	M
	100833	22 May 2023	VIM	Bronchial secretions	ICU A	70	F
	106450	30 May 2023	NDM	Ocular swab	OPHT	41	F
	106563	30 May 2023	VIM	CVC	INT A	58	M
	106902	31 May 2023	VIM	Urine	INT B	94	F
June 2023	111031	6 June 2023	VIM	Blood	INT A	85	F
	110940	6 June 2023	VIM	Blood	INT A	89	F
	113902	11 June 2023	VIM	Blood	BH	66	F
	115521	13 June 2023	NDM	Wound	NS	57	F
	127808	30 June 2023	VIM	Sputum	INT B	57	M
July 2023	133885	1 July 2023	VIM	Urine	INT A	70	M
	129750	4 July 2023	NDM	Rectal surveillance swab	SURGB	70	M
	129929	4 July 2023	VIM	Rectal surveillance swab	ICU A	65	M
	133820	10 July 2023	VIM	Bronchial secretions	ICU A	69	M
	136318	14 July 2023	VIM	Bronchial secretions	ICU A	61	M
	136808	14 July 2023	VIM	Blood	INT B	66	M
	138644	18 July 2023	NDM	Blood	SURGB	43	M
	142123	24 July 2023	VIM	Urine	ICU A	45	M
	144447	27 July 2023	VIM	Sputum	INT B	83	F
August 2023	147614	2 August 2023	VIM	CVC	NS	49	M
	147288	2 August 2023	VIM	BAL	SURGB	50	M
	150146	7 August 2023	VIM	Urine	EMERG	78	M
	151234	8 August 2023	VIM	Urine	NB	82	F
	153153	11 August 2023	ΝDM	Urine	INT A	32	F
	153808	13 August 2023	VIM	Tissue sample	INT A	81	M
	159858	24 August 2023	VIM	Urine	INT A	76	M
	161810	28 August 2023	ΝDM	ΒAL	ICU A	70	F
	162377	29 August 2023	ΝDM	Urine	INT A	81	F
September 2023	164424	2 September 2023	VIM	Urine	INT B	77	F
	165541	4 September 2023	ΝDM	Blood	INT A	76	F
	165760	5 September 2023	VIM	Ocular swab	OPHT	47	M
	168603	8 September 2023	ΝDM	Blood	INT B	85	M
	170177	12 September 2023	VIM	Bronchial secretions	INT B	61	M
	171353	13 September 2023	VIM	Bronchial secretions	SURGB	59	M
	172628	14 September 2023	ΝDM	Urine	INT A	84	F
	175558	19 September 2023	ΝDM	Urine	INT A	76	M
	177569	21 September 2023	VIM	Urine	INT A	61	F
	176948	21 September 2023	ΝDM	Urine	INT A	88	F
	176876	21 September 2023	ΝDM	CVC	INT A	86	F
October 2023	184504	2 October 2023	ΝDM	BAL	ICU A	67	F
	184483	2 October 2023	ΝDM	Urine	ICU A	71	M
	186043	3 October 2023	ΝDM	Blood	INT B	77	F
	186274	4 October 2023	VIM	Bronchial secretions	SURGB	71	M
	188259	6 October 2023	VIM	Wound	INT B	50	F
	188453	6 October 2023	ΝDM	Urine	INT A	69	M
	188456	6 October 2023	ΝDM	Urine	INT B	86	F
	191530	11 October 2023	ΝDM	Urine	ICU A	48	M
	196558	18 October 2023	VIM	Urine	ICU A	55	F
	204219	30 October 2023	ΝDM	Bronchial secretions	INT B	88	M
	205032	31 October 2023	ΝDM	Urine	INT A	88	F
November 2023	206389	2 November 2023	ΝDM	Wound	BH	76	M
	207607	4 November 2023	ΝDM	Urine	INT A	70	F
	207721	4 November 2023	ΝDM	Urine	INT B	73	F
	207883	5 November 2023	ΝDM	Wound	NS	33	F
	208276	6 November 2023	VIM	Rectal surveillance swab	ICU A	46	M
	208520	7 November 2023	VIM	CVC	INT B	87	F
	208299	11 November 2023	VIM	Urine	INT A	80	F
	209832	12 November 2023	ΝDM	Urine	INT B	87	M
	212710	14 November 2023	ΝDM	Rectal surveillance swab	SURGB	80	M
	212971	21 November 2023	ΝDM	Urine	NS	56	M
	214397	22 November 2023	VIM	BAL	SURGB	78	M
	215548	24 November 2023	ΝDM	BAL	ICU A	61	M
	214929	25 November 2023	VIM and NDM	Wound	CARD	70	F
	214898	28 November 2023	VIM	Blood	INT B	87	F
	219774	29 November 2023	ΝDM	Urine	INT B	81	F
December 2023	229861	6 December 2023	VIM	BAL	SURGB	79	M
	231056	7 December 2023	ΝDM	Urine	CARD	92	M
	231677	9 December 2023	ΝDM	Urine	INT A	81	F
	232043	9 December 2023	ΝDM	Wound	CARDSURG	57	F
	232474	10 December 2023	ΝDM	Blood	ICU A	43	M
	236598	15 December 2023	VIM	Urine	INT B	86	M
	238830	19 December 2023	ΝDM	Urine	NEPHR	80	F
	240501	21 December 2023	VIM	Blood	BH	13	M
	242396	25 December 2023	VIM	CVC	ICU A	67	F
	242607	25 December 2023	ΝDM	Urine	INT A	73	F
	243772	27 December 2023	ΝDM	Rectal surveillance swab	INT A	86	F
	245847	31 December 2023	ΝDM	Urine	INT B	85	F
January 2024	830	2 January 2024	VIM	Urine	EMERG	93	F
	71	2 January 2024	ΝDM	Urine	INT B	62	F
	590	2 January 2024	ΝDM	Rectal surveillance swab	INT B	98	M
	954	2 January 2024	ΝDM	Urine	NEPHR	80	F
	8683	15 January 2024	ΝDM	Blood	INT B	82	M
	8653	15 January 2024	ΝDM	Sputum	INT B	67	F
	11691	18 January 2024	ΝDM	Blood	INT A	93	F
	12093	19 January 2024	ΝDM	Blood	INT A	88	M
	13707	22 January 2024	ΝDM	Groin surveillance swab	ICU A	52	M
	15141	23 January 2024	VIM	Rectal surveillance swab	INT B	89	F
	15194	23 January 2024	VIM	Rectal surveillance swab	INT B	71	M
	15253	23 January 2024	ΝDM	Groin surveillance swab	INT B	72	F
	15186	23 January 2024	VIM	Urine	INT A	78	F
	14463	23 January 2024	VIM	Urine	INT A	87	F
	15506	24 January 2024	VIM	BAL	SURGB	45	M
	16012	24 January 2024	VIM	Rectal surveillance swab	CARD	54	F
	20811	31 January 2024	ΝDM	Groin surveillance swab	INT B	85	M
	20810	31 January 2024	ΝDM	Groin surveillance swab	INT B	80	F
February 2024	21862	1 February 2024	VIM	Cerebrospinal fluid	ICU B	54	M
	23947	4 February 2024	NDM	Bronchial secretions	ICU B	87	M
	24299	5 February 2024	NDM	Urine	INT A	89	F
	23926	5 February 2024	VIM	BAL	ICU B	60	M
	26299	8 February 2024	VIM	Urine	INT B	74	M
	26661	8 February 2024	NDM	Groin surveillance swab	INT B	74	M
	30608	14 February 2024	NDM	Groin surveillance swab	INT A	78	M
	30606	14 February 2024	NDM	Rectal surveillance swab	INT A	80	F
	31647	15 February 2024	NDM	Rectal surveillance swab	ICU B	72	M
	32632	16 February 2024	VIM	Rectal surveillance swab	INT B	95	F
	32639	16 February 2024	NDM	Groin surveillance swab	INT B	84	M
	32636	16 February 2024	NDM	Groin surveillance swab	INT B	87	M
	33016	17 February 2024	NDM	Blood	INT B	80	F
	35019	20 February 2024	NDM	Rectal surveillance swab	INT A	40	M
	38019	24 February 2024	NDM	Urine	INT B	67	M
	39050	26 February 2024	VIM	Rectal surveillance swab	ICU B	77	M
	38901	26 February 2024	NDM	Rectal surveillance swab	INT A	79	M
	39189	26 February 2024	NDM	Groin surveillance swab	INT B	86	F
	39177	26 February 2024	NDM	Groin surveillance swab	INT B	78	F
	41326	28 February 2024	VIM	Rectal surveillance swab	INT A	76	M
	41336	28 February 2024	NDM	Groin surveillance swab	INT B	86	M
	41257	28 February 2024	NDM	Groin surveillance swab	INT A	75	F
	40486	28 February 2024	VIM	Sputum	INT B	63	M

**Table 2 antibiotics-13-00762-t002:** Hybrispot results reported for eight *P. aeruginosa* isolates. VIM: Verona integron-encoded metallo-β-lactamase; NDM: New Delhi metallo-β-lactamase.

Isolate ID	Hybrispot Result	NG-Test CARBA 5 Result
11102	*bla* _VIM_ *, sul-1, mut gyrp-T831*	VIM
91845	*bla* _NDM_ *, sul-1, sul-2, aac(6′)-Ib, mut gyrp-T831*	NDM
127808	*bla* _VIM_ *, sul-1, aac(6′)-Ib, mut gyrp-T831*	VIM
129750	*bla* _NDM_ *, sul-1, sul-2, aac(6′)-Ib, mut gyrp-T831*	NDM
138644	*bla* _NDM_ *, sul-1, sul-2, aac(6′)-Ib, mut gyrp-T831*	NDM
165541	*bla* _NDM_ *, sul-1, sul-2, aac(6′)-Ib, mut gyrp-T831*	NDM
171353	*bla* _VIM_ *, sul-1, sul-2, mut gyrp-T831*	VIM
214929	*bla* _NDM_ *, bla_VIM_, sul-1*	NDM, VIM

**Table 3 antibiotics-13-00762-t003:** Antimicrobial resistance and virulence genes found in *P. aeruginosa* isolate 91845.

Isolate	Antimicrobial Resistance Genes	Virulence Genes
91845	*aac(3)-Id* *aac(6′)-Ib-cr* *aac(6′)-Ib3* *aac(6′)-Il* *aadA11* *aph(3″)-Ib* *aph(3′)-IIb* *aph(6)-Id* *bla* _NDM-1_ *bla* _OXA-10_ *bla* _OXA-488_ *bla* _PAO_ *catB7* *crpP* *dfrB5* *floR* *fosA* *msr(E)* *qacE* *qnrVC1* *rmtF* *sul1* *sul2*	*xcpP* *exoU* *flgC* *pchB* *mbtH-like* *lasI* *pilG* *pscS* *pscF* *hcp1*

## Data Availability

The data presented in this study are available in the article.

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
