# Peer review of "Rapid Reversal of Carbapenemase-Producing Pseudomonas aeruginosa Epidemiology from blaVIM- to blaNDM-harbouring Isolates in a Greek Tertiary Care Hospital"

_antibiotics, 2024, doi:10.3390/antibiotics13080762_

Round 1

Reviewer 1 Report

Comments and Suggestions for Authors

Manuscript ID: antibiotics-3108497

Title: Rapid reversal of carbapenemase-producing Pseudomonas aeruginosa epidemiology from blaVIM- to blaNDM-harbouring isolates in a Greek tertiary care hospital

The manuscript discusses the changing epidemiology of carbapenemase-producing Pseudomonas aeruginosa strains, specifically focusing on the shift from VIM-type to NDM-1-type carbapenemases in a hospital setting. It details the detection methods used, including immunochromatographic and molecular assays, and highlights the genomic characterization of an NDM-producing isolate belonging to the high-risk clone ST308. The manuscript emphasizes the implications of these findings for treatment strategies in healthcare institutions. This is the most important result of the manuscript.

However, during the revision of this manuscript specific aspects and questions arose which need some further refinement.

The abstract should clearly state the study's objective and highlight the results that address this objective. Furthermore, I have  doubts whether the results presented here support the conclusion described.

All in all, this part of the methodology and results section urgently needs improvement and broadening of the experimental approach.

Explain in the manuscript why only a single isolate was sequenced. Representative genomic insights are important.

The manuscript described the clinical data. However, correlating molecular findings with clinical data needs improvement. Patient outcomes, duration of hospital stay, and comorbidities were not described. Comparing these data is important for understanding the clinical relevance, genomic findings, and epidemiological data. 

The results section should be improved. The structure needs to clearly outline genomic findings and their implications, ensuring coherence and readability for the reader.

The Manuscript requires English revision, to improve the quality of the language.

Comments on the Quality of English Language

The Manuscript requires English revision, to improve the quality of the language.

Reviewer 2 Report

Comments and Suggestions for Authors

The spread of NDM-producing strains of P. aeruginosa that is occurring in Greece certainly merits the implementation of a thorough epidemiological investigation. From experience, I can recommend here, among others, the PFGE method, which, although developed a long time ago, is still a very useful tool in the epidemiological control of nosocomial infections.

This study raises a very important and timely topic of health medicine. It shows how quickly dangerous multidrug-resistant Gram-negative environmental bacteria spread and thus pose a huge threat to hospitalized patients and often limit therapeutic options to a minimum. I recommend the paper for publication.

Author Response

Reviewer 2

The spread of NDM-producing strains of P. aeruginosa that is occurring in Greece certainly merits the implementation of a thorough epidemiological investigation. From experience, I can recommend here, among others, the PFGE method, which, although developed a long time ago, is still a very useful tool in the epidemiological control of nosocomial infections.

This study raises a very important and timely topic of health medicine. It shows how quickly dangerous multidrug-resistant Gram-negative environmental bacteria spread and thus pose a huge threat to hospitalized patients and often limit therapeutic options to a minimum. I recommend the paper for publication.

-We are grateful to the reviewer for his/her favorable evaluation of our manuscript. The PFGE method is not yet available in our lab. Following the reviewer’s advise, we will consider it for probable future research studies in our hospital.

Reviewer 3 Report

Comments and Suggestions for Authors

The work is very interesting. As the authors point out, the study has some limitations that seem important to me at this level. Indeed, it would be very important, instead of sequencing a strain completely, in the absence of performing some sequencing, to do MLST genomic typing to understand whether it is the same strain or a new strain. As such, it is necessary to complete the study with an epidemiological survey on patients infected before and after the isolation of the first strain (May 8, 2023) and show how there were probably cross-infections that allowed this strain to spread. The use of the term "reversal" here is based on observations of facts: isolation of a new strain. The authors must specify that there was no genetic reversion with the MLST results.

Author Response

Reviewer 3

The work is very interesting. As the authors point out, the study has some limitations that seem important to me at this level. Indeed, it would be very important, instead of sequencing a strain completely, in the absence of performing some sequencing, to do MLST genomic typing to understand whether it is the same strain or a new strain. As such, it is necessary to complete the study with an epidemiological survey on patients infected before and after the isolation of the first strain (May 8, 2023) and show how there were probably cross-infections that allowed this strain to spread. The use of the term "reversal" here is based on observations of facts: isolation of a new strain. The authors must specify that there was no genetic reversion with the MLST results.

-We are grateful to the reviewer for taking the time to review our manuscript and the interest in our work. We appreciate his/her constructive comments, and we regret to not be able to fulfill them at this point. Certain limitations of the study such as lack of MLST for all or for a significant number of isolates, was due to economic restrictions. It would be indeed an option to perform MLST only instead of sequencing, but this would eliminate access to information like carbapenemase-encoding gene location, plasmids and additional resistance mechanisms. In any case, this choice could not change at this stage. The aim of our manuscript was to showcase the reversal of the carbapenemase type encoded by P. aeruginosa in our area after many years of VIM predominance as we were very impressed by how fast this phenomenon is evolving. Retrieving additional patient data, would unfortunately be beyond the reach of our research team. Moreover, patient consent and additional ethical approvement would be probably also needed whereas this is a retrospective study. We apologize for not applying these very interesting suggestions but this is because of practical reasons only.

Round 2

Reviewer 3 Report

Comments and Suggestions for Authors

In the absence of costly molecular typing, this work deserves to be supplemented by descriptive epidemiological data. With a table showing in a first step the most concerned and affected services at the expense of other services not concerned by the appearance of the new strain, Pseudomonas aeruginosa NDM, including the chronological order. In the second step, it is important to include the crossing of infected patients with first the index patient and possibly its previous circulation outside or inside the hospital (first carrier patient). These data are in the files and do not affect the ethics of the affected patients. The tracing of the crossings between the carrier patients can provide interesting data to this manuscript in the absence of molecular typing.

Author Response

Reviewer 3

In the absence of costly molecular typing, this work deserves to be supplemented by descriptive epidemiological data. With a table showing in a first step the most concerned and affected services at the expense of other services not concerned by the appearance of the new strain, Pseudomonas aeruginosa NDM, including the chronological order. In the second step, it is important to include the crossing of infected patients with first the index patient and possibly its previous circulation outside or inside the hospital (first carrier patient). These data are in the files and do not affect the ethics of the affected patients. The tracing of the crossings between the carrier patients can provide interesting data to this manuscript in the absence of molecular typing.

-We would like to thank once again the reviewer for his/her interest in our work and his/her constructive comments. Our data suggest that the departments that were first affected by NDM producing Pseudomonas aeruginosa, were in chronological order, INT A, OPHT, NS and SURGB. These departments are far from one another in the hospital, though connection by moving personnel like physiotherapy, cannot be excluded neither verified retrospectively. Additionally, our hospital is a tertiary care hospital admitting patients from numerous acute- and long-term care facilities and as our colleagues from Larisa suggested, NDM producing Pseudomonas aeruginosa, had already been circulating in northern Greece. That being said, we expect to have had numerous index patients in our hospital. Considering the above, it does not seem safe to draw conclusions upon intrahospital dissemination. In order to address at some extend the comment of the reviewer, we added this information in our discussion section as follows: “Our data suggest that the departments that were first affected by NDM-producing P. aeruginosa, were in chronological order, the internal medicine department A, ophthalmology, neurosurgery and surgery department B. These departments are located far from one another inside the hospital, nevertheless, connection between them (by moving personnel for e.g., physiotherapy) cannot be excluded neither verified retrospectively. Additionally, our institution is a tertiary care hospital admitting patients from numerous acute- and long-term care facilities and as our colleagues from Larisa, Greece suggested, NDM-producing P. aeruginosa, was already circulating in northern Greece [34]. That being said, we expect to have had multiple index patients therefore, it would not seem safe to draw conclusions upon intrahospital dissemination”

This addition is highlighted in blue to facilitate his/her revision.

Round 3

Reviewer 3 Report

Comments and Suggestions for Authors

Thank you for the authors.